# Selenium and Iodine Biofortification Interacting with Supplementary Blue Light to Enhance the Growth Characteristics, Pigments, Trigonelline and Seed Yield of Fenugreek (*Trigonella foenum-gracum* L.)

Sadrollah Ramezani [1], Behnaz Yousefshahi [2], Yusuf Farrokhzad [3], Dariush Ramezan [4,*], Meisam Zargar [5,*] and Elena Pakina [5]

1. Department of Medicinal Plants, Faculty of Geography and Environmental Planning, University of Sistan and Baluchestan, Zahedan 98167-45845, Iran; sramezan@eco.usb.ac.ir
2. Department of Soil Science, Faculty of Agriculture, Gorgan University of Agricultural Sciences and Natural Resources, Gorgan 49189-43464, Iran; yousefshahi@gau.ac.ir
3. Department of Horticultural Science, Faculty of Agriculture, Tarbiat Modares University, Tehran 14117-13116, Iran; farrokhzadyusuf@gmail.com
4. Department of Horticulture and Landscaping, Faculty of Agriculture, University of Zabol, Zabol 98613-35856, Iran
5. Department of Agrobiotechnology, Institute of Agriculture, RUDN University, 117198 Moscow, Russia; e-pakina@yandex.ru
* Correspondence: drhorticul@uoz.ac.ir (D.R.); zargar_m@pfur.ru (M.Z.)

**Abstract:** Fenugreek (*Trigonella foenum-graecum*) is an annual plant belonging to the family Fabaceae and has fodder, medicinal and spice uses, and is also used as an organic fertilizer. A total of 18 treatments including the combination of two light environments (with and without supplementary blue light), three concentrations of potassium iodate (0, 2 and 4 mg L$^{-1}$) and four concentrations of sodium selenate (0, 2 and 4 mg L$^{-1}$) were organized in a three-way factorial experiment to evaluate the growth characteristics, pigments, trigonelline and seed yield of fenugreek in a greenhouse. The application of 4 mg L$^{-1}$ of Se resulted in the highest carotenoid, anthocyanin, plant length, fresh weight, chlorophyll and relative water content. The fresh and dry weight of the shoot and the anthocyanin increased with the 2 h supplementation of sunlight with a blue spectrum; however, the fresh root decreased. The interaction of blue light with 0 mg L$^{-1}$ of Se significantly reduced the plant length. The content of trigonelline was significantly improved with the application of blue light supplementation without negatively affecting the seed yield. In general, 2 h supplementing of sunlight with blue light and feeding with 4 mg L$^{-1}$ of selenium and iodine are recommended to improve various traits, including trigonelline content.

**Keywords:** biofortification; blue supplementary lighting; growth analysis; trigonelline content; plant pigments

## 1. Introduction

Fenugreek (*Trigonella graecum-foenum* L.) is an annual plant and belongs to the legume family, which has fodder, medicinal and spice uses, is also used as an organic fertilizer and produces about five tons per hectare of a relatively significant biomass [1]. The seeds of the plant have nutritional properties and stimulate the digestion process [2]. Fenugreek is known as one of the oldest medicinal plants cultivated in many countries, especially India, North Africa, the Mediterranean and Canada [3]. Fenugreek is a good source of proteins; soluble and insoluble dietary fibers; crude fats; minerals such as calcium, iron and β-carotene [4]; phospholipids; glycolipids; oleic acid; linolenic acid; linoleic acid [5]; choline; vitamins A, B1, B2 and C; nicotinic acid; and niacin [6], and because of this, it

has remarkable biological activities including protection against cancer, malaria, allergies, bacteria and viruses [7]. Fenugreek contains compounds such as saponin diosgenin and pyridine alkaloid trigonelline, and especially the last one may play a role in the treatment of many diseases such as diabetes, due to its hypoglycemic, neuroprotective, anti-invasive, estrogenic and antibacterial properties [8].

Trace elements like iodine (I) and selenium (Se) are vital for the normal function of the thyroid gland [9]. The simultaneous biofortification of crops with I and selenium is of great interest for lands with a deficiency of these two elements in the soils, because I and Se are essential nutrients for humans. The main justification for the deficiency of both elements in the human diet is the inadequate intake of I and Se from vegetables [10,11].

Se is an essential micronutrient element that plays a vital role in various biological processes such as thyroid hormone metabolism, the antioxidant defense system and also improving the function of the immune system [12]. Furthermore, Se is involved in the development and function of living cells [13]. Epidemiological studies have confirmed that Se deficiencies in the diet increase the risk of cardiovascular disease and malfunction of the thyroid gland, as well as the immune and nervous systems [14]. I is also a micronutrient that represents an important role in the right physiological function of mammals such as humans and animals [15]. The World Health Organization (WHO) has identified an I deficiency as one of the main factors affecting human health [16]. I deficiency disorders are the insufficient production of thyroid hormones, which results in goiter [17].

The enrichment of agricultural crops with Se aims to improve the nutritional value of crops for humans [18]. Se also has a significant influence on the growth and quality of vegetables. Se application leads to a significant increase in Se content in the foliage of spinach (*Spinacia oleracea*) [19], broccoli (*Brassica oleracea* var. italica) [20], cabbage (*Brassica oleracea*) [21] and radish (*Raphanus sativus*) [22], without undesirable effects on the quantitative and qualitative yields of crops. Research conducted by He et al. [23] demonstrated that the content of carotenoid, total phenol, total flavonoid, total carbohydrate and total protein, nutritional value, dry weight, Se and mineral amounts (potassium, calcium and zinc) of edible sprouts of broccoli increased meaningfully under the application of Se ($100 \, \mu \, \mathrm{mol} \, \mathrm{L}^{-1}$) and light quality (blue, red and green) combinations. Previous studies showed that total chlorophyll values of lettuce (*Lactuca sativa*) [24] and broccoli [25] increased via Se application. In another study, Se application reduced the chlorophyll content of mint (*Mentha* × *piperita*) while increasing the carotenoids [26]. Se has also been reported to increase total carbohydrates in tomato fruit [27]. Gonnella et al. [28] also showed that in the hydroponic cultivation of cabbage, incorporating I into the nutrient solution increases the dry weight and I content in the cabbage foliage.

Recently, light-emitting diode (LED) technology has provided the feasibility of regulating the growth and development of plants by modifying the spectral composition of light [29]. The light parameters caused with LED, such as the quality, intensity and duration of light, affect plant morphology and function by inducing responses at the biochemical, physiological and morphological levels, and also improve the nutritional properties of fruits and vegetables by affecting bioactive compounds [30]. In various studies, light in the blue region has been used mainly to improve the rate of photosynthesis and phytochemicals. A study by Son and Oh [31] concluded that blue light increases the total phenol and antioxidant capacity of red lettuce leaves. The total phenol and flavonoid content and antioxidant capacity of lettuce grown under higher ratios of blue light were significantly higher compared to other spectra [32]. Shin et al. [33] also reported that the accumulation of calcium, magnesium, manganese and iron in lettuce foliage was induced with blue light. Terfa et al. [34] showed that the increase in blue light from 5 to 20% increases the thickness of the leaf and therefore results in increased photosynthetic activity.

We hypothesized that supplementary blue lighting at the end of the morning (for 2 h) can increase the content of chemicals, especially trigonelline in fenugreek foliage without delaying growth and development processes. Also, our other hypothesis was that, as a result of feeding plants with I and Se in addition to increasing the content of these two

useful elements for human nutrition, the quantitative and qualitative yield of the crop also increases. Therefore, the aim of this study was to evaluate the mechanisms of the growth response, quantitative and qualitative yield and biochemical content (trigonelline content) of the fenugreek plant regarding blue light and nutrition with Se and I through measuring the fresh and dry weight of plants, plant pigments, seed yield and trigonelline content.

## 2. Materials and Methods

### 2.1. Growing Conditions

This research was laid out as a three-factor factorial experiment in a completely randomized design (CRD) with three replications in pots in the research greenhouse of the Faculty of Agriculture of Zabol University in 2022. In this experiment, the first factor was three levels of a sodium selenate fertilizer (0, 2 and 4 mg L$^{-1}$), the second factor was three levels of potassium iodate (0, 2 and 4 mg L$^{-1}$) and the third factor was two lighting conditions: supplementing with blue light (450–520 nm) and no supplementation with blue light. To provide supplemental blue light, LED chips were purchased from Iran Grow Light Company (Tehran, Iran). Fenugreek seeds were purchased from the Pakan Bazar company (Isfahan, Iran). After washing, the seeds were rinsed three times with distilled water and then treated with a benomyl fungicide (2 g L$^{-1}$ of water) for 20 min. After disinfection, 10 seeds were planted in each pot (4 L) on a substrate of cocopeat/perlite (1:1). After germination, thinning was performed whereby five seedlings were kept and the rest were removed. Fertilizer application using a Hoagland's nutrient solution supplemented with sodium selenate ($Na_2SeO_4$) and potassium iodate ($KIO_3$) was performed. The greenhouse is located in the north–south direction; has a polycarbonate cover; is equipped with a cooling device, lateral ventilation and shade; has a mean daily temperature of 22–28 °C and relative humidity of 60–70%; and receives 600 μmol m$^{-2}$ s$^{-1}$ light. After the establishment of the seedlings and the emergence of four true leaves (20 days after sowing), the irradiation of 15 μmol m$^{-2}$ s$^{-1}$ of blue light started. The light intensity was tailored by adjusting the distance of the lamps from the plants and from each other. An analog timer was used to set the lighting for 2 h at dawn.

### 2.2. Morphological Traits

Plant height was measured with a ruler, and the fresh weight of the shoot, root and seed was measured with a digital scale. Also, the aerial parts and the root of each plant were placed in an oven separately for 72 h at 65 °C, and then the dried samples were weighed to obtain the dry weight.

### 2.3. Physiological and Biochemical Traits

#### 2.3.1. Leaf Relative Water Content (RWC)

First, the samples were placed in distilled water and kept at a temperature of 4 °C for 24 h. After 24 h, the saturated weight of the leaves was measured and the leaves were placed in the oven at 70 °C for 24 h and the dry weight was estimated. Then, RWC was determined by putting the obtained numbers in the following formula [35]:

$$RWC = Fw - Dw/Sw - Dw \times 100$$

Fw: Leaf fresh weight.
Dw: Leaf dry weight.
Sw: Leaf saturated weight.

#### 2.3.2. Total Chlorophyll and Carotenoids

Fresh aerial samples of 0.2 g were placed in 8 mL of an ethanol–acetone mixture (1:1) at 25 °C for 24 h and incubated in a dark place until the appearance of a white color. The absorbance of the solution was read using a spectrophotometer at wavelengths of 645, 663

and 440 nm. Chlorophyll and carotenoid quantification was conducted using the following equations [36]:

$$\text{Chlorophyll } a \text{ (mg L}^{-1}) = 12.7 \times \text{A663} - 2.69 \times \text{A645}$$

$$\text{Chlorophyll } b \text{ (mg L}^{-1}) = 22.9 \times \text{A645} - 4.86 \times \text{A663}$$

$$\text{Total chlorophyll (mg L}^{-1}) = 8.02 \times \text{A663} + 20.20 \times \text{A645}$$

$$\text{Total carotenoids (mg L}^{-1}) = 4.7 \times \text{A440} - 0.27 \times \text{total chlorophyll}$$

### 2.3.3. Total Anthocyanin

Total anthocyanin content was measured according to the method described by Xu et al. [37]. Plant samples were heated with 20 mL of 60% ethanol for 2 h in a hot water bath, then the samples were filtered. The extract solution was read spectrophotometrically at 535 nm. After 15 min, the solution was read at 705 nm [38].

### 2.3.4. Trigonelline Content of Shoot and Seed

A spectrophotometric method was used to determine the amount of trigonelline. In total, 1 g of the plant samples (leaf or seed) was mixed with 1 g of magnesium oxide and 20 mL of distilled water and placed in a hot water bath at 100 °C for 18 min. After cooling, the sample was filtered using filter paper and the volume of the samples was increased to 25 mL by adding distilled water. The samples were centrifuged at 1200 rpm. Next, the supernatant solution was read spectrophotometrically at 268 nm. Using the standard curve, the trigonelline amount was reported as mg per 100 g of the fresh weight of aerial parts and mg per g of the dry weight of seeds [39].

### *2.4. Data Analysis*

The obtained data were statistically analyzed with SAS 9.4 software (SAS Institute Inc., Cary, NC, USA) and mean comparison was conducted with an LSD test at the 0.05 significance level, and cutting was performed based on I levels.

## 3. Results

### *3.1. Plant Growth and Seed Yield*

### 3.1.1. Plant Length

As shown in Table 1, under the conditions of supplementary blue light, the highest (15.77 cm) plant length was observed at 4 mg L$^{-1}$ of Se, which is not significantly different from 2 mg L$^{-1}$ treatment in the vegetative state (40 days after sowing). Also, under blue light, the lowest plant length (12.44 cm) was recorded in untreated Se (0 mg L$^{-1}$) in the vegetative stage. The length of plants in the vegetative stage with sunlight (control) in all three levels of Se was in the same statistical group. In the pod formation stage (80 days after sowing), the highest (34.37 cm) and the lowest (29.50 cm) plant lengths were found in the non-supplemented blue light (sunlight background) and supplemented blue light conditions, respectively (Table 2). Also, the highest (35.50 cm) and lowest (28.50 cm) plant lengths in the pod formation stage were noticed in 4 and 0 mg L$^{-1}$ of Se, respectively (Table 3).

**Table 1.** Mean comparison of light and selenium combined effects on plant height, chlorophyll and RWC in fenugreek in 40 days after seed planting.

| Light | Selenium | Plant Height (cm) | Chlorophyll (Mg·g FW) | RWC (%) |
|-------|----------|-------------------|------------------------|---------|
| Blue | 0 | 12.44 ± 1.1 b | 3.84 ± 0.2 c | 64.88 ± 1.9 c |
| | 2 | 15.77 ± 1.1 a | 4.76 ± 0.2 b | 65.11 ± 2.2 b |
| | 4 | 15.00 ± 1.0 a | 5.85 ± 0.2 a | 72.66 ± 3.3 a |

**Table 1.** *Cont.*

| Light | Selenium | Plant Height (cm) | Chlorophyll (Mg·g FW) | RWC (%) |
|---|---|---|---|---|
| | 0 | 18.66 ± 1.2 a | 2.71 ± 0.3 c | 64.55 ± 2.4 c |
| Control | 2 | 18.22 ± 1.1 a | 3.39 ± 0.2 b | 61.55 ± 2.7 b |
| | 4 | 19.44 ± 1.0 a | 4.20 ± 0.2 a | 65.55 ± 2.2 a |
| LSD | | 1.3 | 0.2 | 4.0 |

Data are presented as treatment means ± SE (n = 3). Different letters indicate significantly different values at $p < 0.05$.

**Table 2.** Mean comparison of effect of light on plant height, fresh weight of shoot, RWC, chlorophyll and anthocyanin in fenugreek plant in 80 days after seed planting.

| Light | Plant Height (cm) | FWS (gr) | RWC (%) | Chlorophyll (Mg·g FW) | Anthocyanin (Ug·g FW) |
|---|---|---|---|---|---|
| Blue | 29.50 ± 2.0 b | 75.99 ± 2.6 a | 54.07 ± 2.8 a | 3.49 ± 0.3 a | 8.10 ± 0.5 a |
| Control | 34.37 ± 2.2 a | 69.20 ± 2.5 b | 46.63 ± 3.6 b | 2.69 ± 0.3 b | 6.92 ± 0.5 b |
| LSD | 1.3 | 1.8 | 1.8 | 0.2 | 0.3 |

FWS: Fresh weight of shoot. Data are presented as treatment means ± SE (n = 3). Different letters indicate significantly different values at $p < 0.05$.

**Table 3.** Mean comparison of effect of selenium on carotenoids, anthocyanin, plant height, fresh weight of shoot, chlorophyll and RWC in fenugreek plant.

| Selenium | Carotenoids (Mg·g FW) | Anthocyanin (Ug·g FW) | Plant Height (cm) | FWS (g) | Chlorophyll (Mg·g FW) | RWC (%) | Anthocyanin (Ug·g FW) |
|---|---|---|---|---|---|---|---|
| | $DAP_{40}$ | | | $DAP_{80}$ | | | |
| 0 | 0.34 ± 0.04 c | 3.68 ± 0.2 c | 28.50 ± 1.7 c | 69.50 ± 2.3 b | 2.67 ± 0.3 c | 44.16 ± 3.5 b | 6.54 ± 0.4 c |
| 2 | 0.41 ± 0.04 b | 5.46 ± 0.3 b | 31.72 ± 2.1 b | 71.24 ± 2.9 b | 3.02 ± 0.3 b | 52.66 ± 2.4 a | 7.63 ± 0.4 b |
| 4 | 0.54 ± 0.05 a | 5.85 ± 0.3 a | 35.5 ± 2.0 a | 76.85 ± 3.0 a | 3.59 ± 0.3 a | 54.22 ± 3.0 a | 8.37 ± 0.5 a |
| LSD | 0.03 | 0.2 | 1.5 | 2.1 | 0.2 | 2.2 | 0.4 |

$DAP_{40, 80}$: 40 and 80 days after seed planting, FWS: Fresh weight of shoot. Data are presented as treatment means ± SE (n = 3). Different letters indicate significantly different values at $p < 0.05$.

### 3.1.2. Fresh Weight of Shoots

At the vegetative stage, the fresh weight of shoots (FWS) in the supplemented blue light and non-supplemented blue light conditions was 55.91 g and 50.53 g, respectively (Table 4). Also, no significant differences were detected in 0 mg $L^{-1}$ of Se between 2 and 4 mg $L^{-1}$ of I. The lowest (44.73 g) FWS was found at 0 mg $L^{-1}$ of I. At 2 and 4 mg $L^{-1}$ of Se, there were no significant differences among I treatments in terms of FWS (Table 5). At the pod formation stage, the highest and lowest (75.99 and 69.20 g) FWS were found in the supplemented blue light and non-supplemented blue light conditions, respectively (Table 2). Moreover, no significant difference was found between 0 and 2 mg $L^{-1}$ of Se, and the highest (76.85 g) FWS was detected in 4 mg $L^{-1}$ of Se (Table 3).

**Table 4.** Mean comparison of effect of light on fresh weight of shoot, dry weight of shoot, fresh weight of root and anthocyanin in fenugreek plant in 40 days after seed planting.

| Light | FWS (g) | DWS (g) | FWR (g) | Anthocyanin (Ug·g FW) |
|---|---|---|---|---|
| Blue | 55.91 ± 2.6 a | 11.44 ± 0.5 a | 8.29 ± 0.4 b | 5.24 ± 0.4 a |
| Control | 50.53 ± 2.8 b | 9.55 ± 0.6 b | 11.31 ± 0.5 a | 4.76 ± 0.3 b |
| LSD | 1.7 | 0.5 | 0.4 | 0.2 |

FWS: Fresh weight of shoot, DWS: Dry weight of shoot, FWR: Fresh weight of root. Data are presented as treatment means ± SE (n = 3). Different letters indicate significantly different values at $p < 0.05$.

**Table 5.** Mean comparison of selenium and I combined effects on fresh weight of shoot, dry weight of shoot and chlorophyll in fenugreek in 40 days after seed planting.

| Selenium | I | FWS (g) | DWS (g) | Chlorophyll (Mg·g FW) |
|---|---|---|---|---|
| 0 | 0 | 44.73 ± 3.8 b | 10.25 ± 0.5 ab | 3.11 ± 0.3 ab |
|  | 2 | 53.06 ± 2.0 a | 9.72 ± 0.7 b | 3.27 ± 0.3 a |
|  | 4 | 51.47 ± 1.9 a | 11.30 ± 0.7 a | 2.76 ± 0.2 b |
| 2 | 0 | 54.17 ± 2.0 a | 10.97 ± 0.7 ab | 3.84 ± 0.4 b |
|  | 2 | 52.96 ± 1.9 a | 11.25 ± 0.7 a | 3.84 ± 0.4 b |
|  | 4 | 52.80 ± 1.7 a | 10.15 ± 0.7 b | 4.54 ± 0.3 a |
| 4 | 0 | 57.63 ± 3.0 a | 10.70 ± 0.7 a | 4.78 ± 0.5 a |
|  | 2 | 56.57 ± 2.0 a | 10.03 ± 0.9 a | 5.12 ± 0.5 a |
|  | 4 | 55.61 ± 3.5 a | 10.12 ± 1.1 a | 5.17 ± 0.6 a |
| LSD |  | 2.1 | 1.1 | 0.45 |

FWS: Fresh weight of shoot, DWS: Dry weight of shoot. Data are presented as treatment means ± SE (n = 3). Different letters indicate significantly different values at $p < 0.05$.

### 3.1.3. Dry Weight of Shoot

At the vegetative stage, the highest (11.44) and the lowest (9.55 g) dry weights of the shoot (DWS) were found in the supplemented blue light and non-supplemented blue light conditions, respectively (Table 4). Also, as shown in Table 5, in the no Se application condition (0 mg $L^{-1}$), the maximum DWS was achieved in 4 mg $L^{-1}$ of I, which was not remarkably different from 0 mg $L^{-1}$ of I. Furthermore, in the 2 mg $L^{-1}$ Se application, the highest (11.25 g) DWS was recorded followed by 0 mg $L^{-1}$ of I. At 4 mg $L^{-1}$ of Se, there was no statistical difference between all I concentrations. At the pod formation stage, DWS was higher under supplemented blue lighting in combination with the 0 mg $L^{-1}$ Se application. In addition, no significant difference was noticed between the 2 and 4 mg $L^{-1}$ Se treatments in the supplemented blue light condition. Also, in non-supplemented blue light conditions, no significant difference was found among Se levels (Table 6).

**Table 6.** Mean comparison of light and selenium combined effects on fresh weight of root and dry weight of shoot and root in fenugreek in 80 days after seed planting.

| Light | Selenium | DWS (g) | FWR (g) | DWR (g) |
|---|---|---|---|---|
| Blue | 0 | 22.78 ± 1.3 a | 11.48 ± 0.4 a | 2.53 ± 0.2 a |
|  | 2 | 20.72 ± 1.7 b | 11.08 ± 0.5 a | 2.42 ± 0.1 a |
|  | 4 | 20.10 ± 0.6 b | 11.34 ± 0.3 a | 1.95 ± 0.2 b |
| Control | 0 | 16.91 ± 0.4 a | 13.56 ± 0.4 b | 3.22 ± 0.2 a |
|  | 2 | 17.53 ± 0.9 a | 13.86 ± 1.0 b | 3.45 ± 0.4 a |
|  | 4 | 17.06 ± 0.7 a | 14.70 ± 1.0 a | 3.27 ± 0.3 a |
| LSD |  | 0.9 | 0.4 | 0.3 |

DWS: Dry weight of shoot, FWR: Fresh weight of root, DWR: Dry weight of root. Data are presented as treatment means ± SE (n = 3). Different letters indicate significantly different values at $p < 0.05$.

### 3.1.4. Fresh Weight of Roots

At the vegetative stage, the highest (11.31) and lowest (8.29 g) fresh weight of roots (FWR) was noticed in non-supplemented blue light and supplemented blue light conditions, respectively (Table 4). At the pod formation stage, FWR was higher in the supplemented blue light conditions. But there was no significant difference between various levels of Se (Table 6). Also, at the pod formation stage, in non-supplemented blue light conditions, the highest FWR (14.7 g) was observed in the 4 mg $L^{-1}$ Se application. Also, there was no significant difference between 0 and 2 mg $L^{-1}$ of Se. As presented in Figure 1, in the pod formation stage, at 0 mg $L^{-1}$ of Se, the highest FWR (13.28 g) was achieved in combination with 0 mg $L^{-1}$ of I, and a significant difference was not observed between the 2 and 4 mg $L^{-1}$ I applications. At 2 mg $L^{-1}$ of Se, there was no significant difference

among I treatments. However, in 4 mg L$^{-1}$ of Se, the highest FWR (14.60 g) was detected at 4 mg L$^{-1}$ of I.

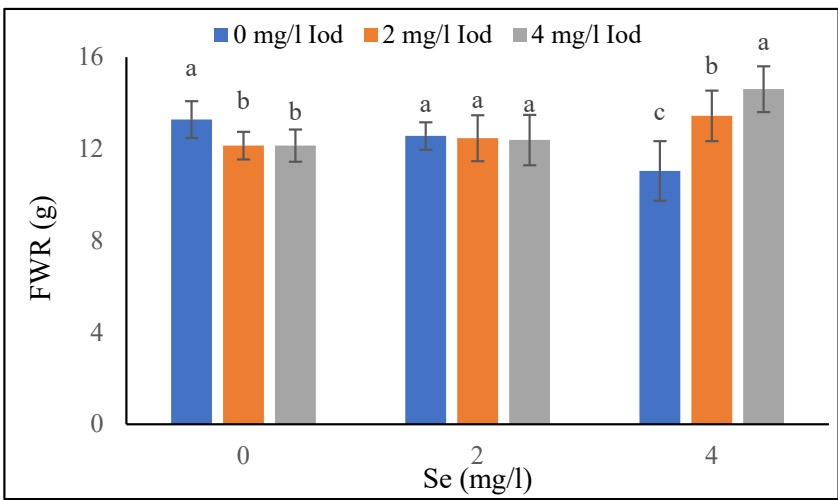

**Figure 1.** Mean comparison of selenium and I combined effects on fresh weight of root in fenugreek in 80 days after seed planting. Different letters indicate significantly different values at *p* < 0.05.

3.1.5. Dry Weight of Roots

At the vegetative stage, the highest dry weight of the root (DWR) under the conditions of supplementary blue light and 0 mg L$^{-1}$ of Se was obtained in 4 mg L$^{-1}$ of I, which was in the same statistical group as 2 mg L$^{-1}$ of I (Figure 2). Under the condition of blue light supplementation, the highest (1.81 g) was detected in 2 mg L$^{-1}$ of I, and it was not significantly different with 4 mg L$^{-1}$ of I. At the pod formation stage, in 4 mg L$^{-1}$ of Se, the highest (3.29) and the lowest (1.93 g) DWR were found in 4 and 0 mg L$^{-1}$ I treatments, respectively (Figure 3).

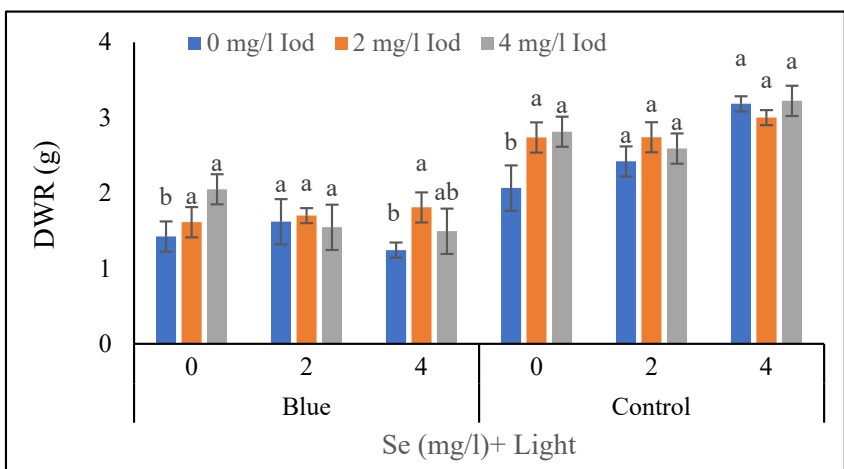

**Figure 2.** Mean comparison of selenium, I and light combined effects on dry weight of root in fenugreek in 40 days after seed planting. Different letters indicate significantly different values at *p* < 0.05.

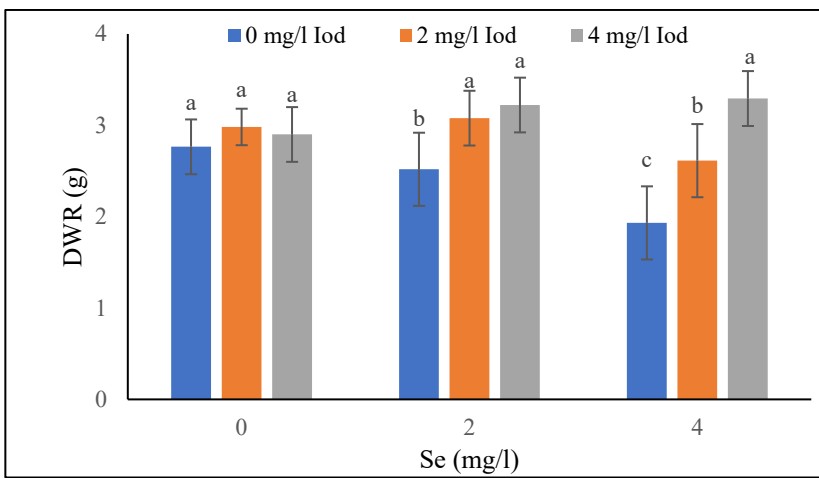

**Figure 3.** Mean comparison of selenium and I combined effects on dry weight of root in fenugreek in 80 days after seed planting. Different letters indicate significantly different values at *p* < 0.05.

### 3.1.6. Relative Water Content (RWC) of Leaves

As shown in Figure 4, in the vegetative stage, the highest RWC (68.33%) of the leaves was recorded in the 0 mg L$^{-1}$ I treatment, and also there was no significant difference between 2 and 4 mg L$^{-1}$ of I. In the supplemented blue light conditions, the highest RWC (72.66%) was matched with the 4 mg L$^{-1}$ Se treatment, and also there was no significant difference between 0 and 2 mg L$^{-1}$ of Se (Table 1). In non-supplemented blue light conditions, there was no significant difference between 0 and 4 mg L$^{-1}$ of Se. At the pod formation stage, the highest (54.07%) and lowest (46.63%) RWC was obtained in the supplemented blue light and non-supplemented blue light conditions, respectively (Table 2). In addition, the highest (54.22%) and lowest (44.16%) RWC was recorded in 4 and 0 mg L$^{-1}$ of Se, respectively (Table 3).

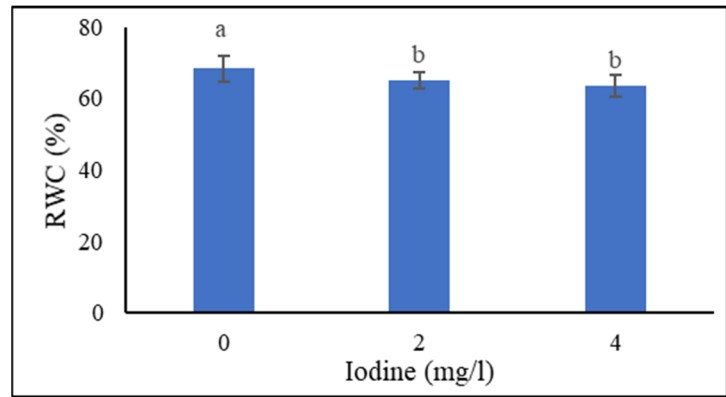

**Figure 4.** Mean comparison of I effect on RWC in fenugreek plant 40 days after seed planting. Different letters indicate significantly different values at *p* < 0.05.

### 3.1.7. Seed Yield

In the combination of 0 mg L$^{-1}$ of Se and supplementary blue light, there was no significant difference among different I treatments (Figure 5). Also, in the supplemented blue light condition and 2 mg L$^{-1}$ of Se, the various I concentrations had no significant influence on seed yield. In the supplemented blue light conditions and 4 mg L$^{-1}$ of Se, the highest (4.48 g plant$^{-1}$) seed yield was found in 0 mg L$^{-1}$ of I and no significant difference was detected between 2 and 4 mg L$^{-1}$ of I. Under non-supplemented blue light conditions with 0 mg L$^{-1}$ of Se, the highest seed yield (2.82 g plant$^{-1}$) was obtained in 2 mg L$^{-1}$ of I and there was no significant difference between 2 and 4 mg L$^{-1}$ of I. In the

non-supplemented blue light conditions with 2 and 4 mg $L^{-1}$ of Se, no difference was observed between 0 and 2 mg $L^{-1}$ of I (<5%) (Figure 5).

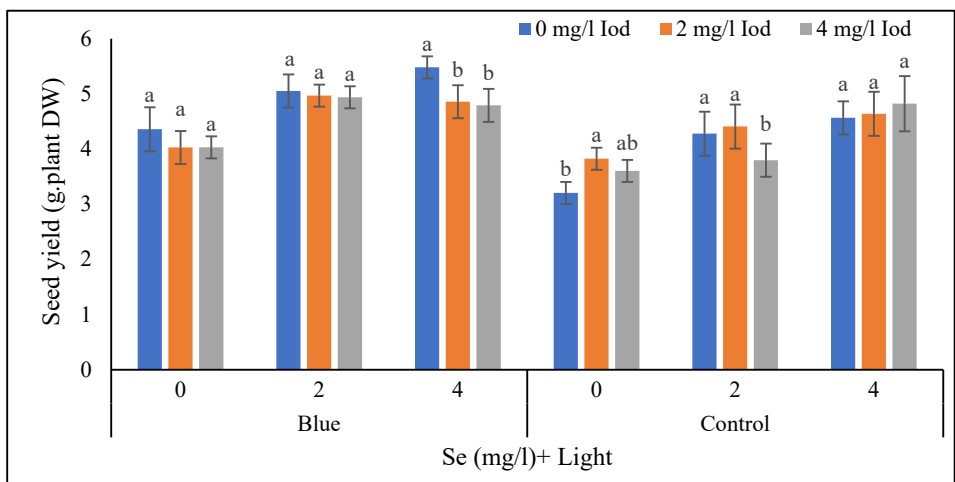

**Figure 5.** Mean comparison of selenium, I and light combined effects on seed yield of fenugreek plant. Different letters indicate significantly different values at *p* < 0.05.

### 3.2. Plant Pigments

3.2.1. Total Chlorophyll

At the vegetative stage, under both lighting environments, the highest and lowest chlorophyll content was noticed in 4 and 0 mg $L^{-1}$ of Se, respectively (Table 1). Also, there was no significant difference between 0 and 2 mg $L^{-1}$ of I in combination with 0 mg $L^{-1}$ of Se (Table 5). Also, in 2 mg $L^{-1}$ of Se, the highest (4.54 mg $g^{-1}$) chlorophyll was observed in 4 mg $L^{-1}$ of I and also no significant difference was observed between I levels in 4 mg $L^{-1}$ of Se. At the pod formation stage, the highest (3.49 mg $g^{-1}$) and lowest (2.69 mg $g^{-1}$) chlorophyll content was obtained in the supplementary blue light condition followed by non-supplemented blue light conditions, respectively (Table 2). Also, the highest (3.59 mg $g^{-1}$) and lowest (2.67 mg $g^{-1}$) chlorophyll content was recorded in 4 and 0 mg $L^{-1}$ of Se, respectively. No significant difference in leaf chlorophyll was found between 0 and 2 mg $L^{-1}$ of I (Figure 6).

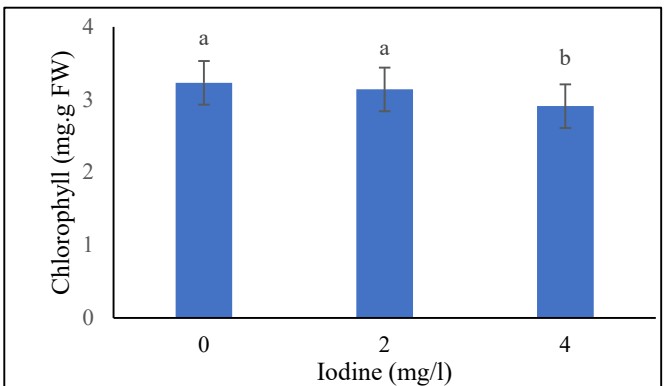

**Figure 6.** Mean comparison of I effect on chlorophyll in fenugreek plant 80 days after seed planting. Different letters indicate significantly different values at *p* < 0.05.

3.2.2. Anthocyanin Content

At vegetative stages, the highest (5.24 mg $g^{-1}$ FW) and lowest (4.76 mg $g^{-1}$ FW) anthocyanin was associated with the supplemented blue light and non-supplemented blue light conditions, respectively (Table 4), and the highest (5.85 mg $g^{-1}$ FW) and the lowest

(3.68 mg g$^{-1}$ FW) anthocyanin was found in 4 and 0 mg L$^{-1}$ of Se, respectively (Table 3). As shown in Table 2, the highest (8.10 mg g$^{-1}$ FW) and the lowest (6.92 mg g$^{-1}$ FW) anthocyanin contents were recorded in the supplementary blue light condition followed by the non-supplemented blue light condition, respectively. At the pod formation stage, as presented in Table 3, the highest and the lowest anthocyanin was recorded in 4 and 0 mg L$^{-1}$ of Se, respectively (Table 3).

### 3.2.3. Carotenoid Content

As shown in Table 3, in the vegetative stage, the highest and the lowest carotenoid amount was found in 4 and 0 mg L$^{-1}$ of Se, respectively. Furthermore, in the supplementary blue light treatment, the highest and the lowest carotenoid amount was noticed in 4 and 0 mg L$^{-1}$ of I, respectively. Under non-supplemented blue light conditions, the highest (0.44 mg g$^{-1}$ FW) and the lowest (0.35 mg g$^{-1}$ FW) carotenoid amount was associated with 4 and 2 mg L$^{-1}$ of I, respectively, and there was no significant difference between the levels of 0 and 2 mg L$^{-1}$ of I (Figure 7). At the pod formation stage, carotenoid contents under the blue light treatment were the highest (0.75 g$^{-1}$ FW) and the lowest (0.58 mg g$^{-1}$ FW) in 4 and 0 mg L$^{-1}$ of I, respectively. In addition, in the supplemented blue light, the highest (0.64 mg g$^{-1}$ FW) and lowest (0.55 mg g$^{-1}$ FW) carotenoid amount corresponded with 4 and 2 mg L$^{-1}$ of I, respectively, and no significant difference was detected between 0 and 2 mg L$^{-1}$ of I (Figure 8).

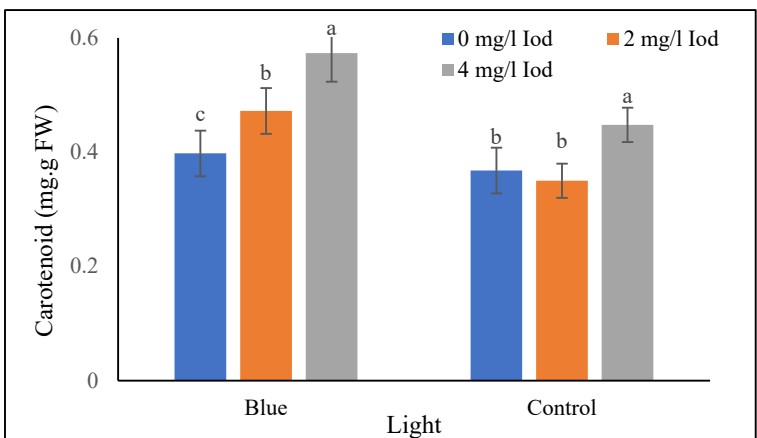

**Figure 7.** Mean comparison of combined effect of light and I on carotenoids in fenugreek in 40 days after seed planting. Different letters indicate significantly different values at *p* < 0.05.

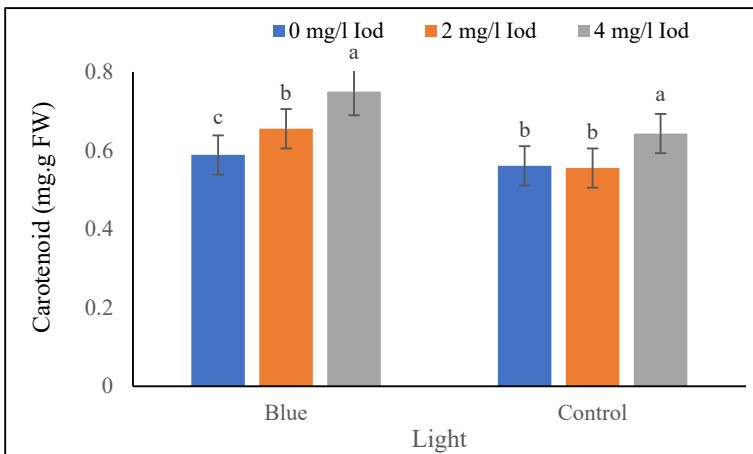

**Figure 8.** Mean comparison of light and I combined effect on carotenoid content in fenugreek in 80 days after seed planting. Different letters indicate significantly different values at *p* < 0.05.

*3.3. Trigonelline Content of Shoot and Seed*

3.3.1. Shoot Trigonelline Content

As seen in Figure 9, in the supplemented blue light condition and 0 mg L$^{-1}$ of Se, the highest and the lowest trigonelline shoot contents were detected in 4 and 2 mg L$^{-1}$ of I, respectively. Moreover, at the vegetative stage in the same light condition and 2 mg L$^{-1}$ of Se, the highest and the lowest amount of trigonelline was recorded in 0 and 2 mg L$^{-1}$ of I, respectively. In the supplemented blue light treatment with 4 mg L$^{-1}$ of Se, the highest and the lowest trigonelline content corresponded with 4 and 0 mg L$^{-1}$ of I, respectively. In contrast, in non-supplemented blue light conditions and 0 mg L$^{-1}$ of Se, the highest and the lowest concentrations of trigonelline were noted in 4 and 0 mg L$^{-1}$ of I, respectively, and no significant difference was detected between 0 and 2 mg L$^{-1}$ of I. Under these lighting conditions with 2 mg L$^{-1}$ of Se, the highest and the lowest content of trigonelline was recorded in 0 and 2 mg L$^{-1}$ of I, respectively, and no significant difference was detected between 2 and 4 mg L$^{-1}$ of I. In addition, in non-supplemented blue light conditions and 4 mg L$^{-1}$ of Se, the highest and the lowest rigonelline content was observed in 4 and 0 mg L$^{-1}$ of I, respectively (Figure 9). As shown in Figure 10, at the pod formation stage and high-blue-lighting condition with 0 mg L$^{-1}$ of I, the highest and the lowest trigonelline content was found in 0 and 2 mg L$^{-1}$ of I, respectively, and no significant differences were found between 2 and 4 mg L$^{-1}$ of I. In these lighting conditions with 2 mg L$^{-1}$ of Se, the highest and the lowest trigonelline amounts were detected in 0 and 4 mg L$^{-1}$ of I. Furthermore, in the supplemented blue light condition with 4 mg L$^{-1}$ of Se, the highest and the lowest trigonelline content was obtained in 2 and 0 mg L$^{-1}$ of I, respectively. On the other hand, in the non-supplemented blue light conditions with 0 mg L$^{-1}$ Se conditions, no significant difference was detected among I levels. Additionally, at the pod formation stage, in the supplemented blue light condition with 2 mg L$^{-1}$ of Se, the highest and the lowest contents of trigonelline were matched to 0 and 2 mg L$^{-1}$ of I, respectively. At the pod formation stage, under the combination of the supplemented blue light condition and 4 mg L$^{-1}$ of Se, the highest and the lowest trigonelline content was noticed in 4 and 0 mg L$^{-1}$ of I, respectively (Figure 10).

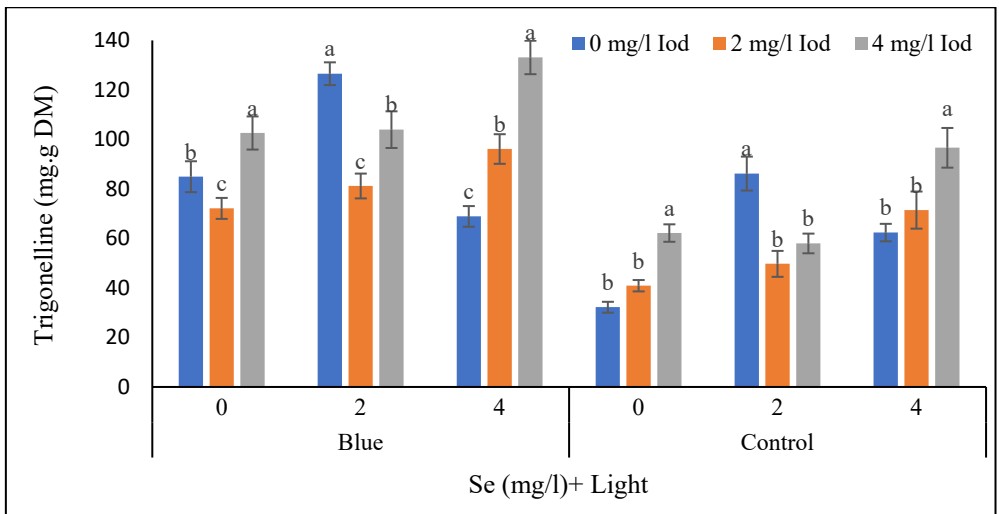

**Figure 9.** Mean comparison of selenium, I and light combined effects on trigonelline in fenugreek shoots in 40 days after seed planting. Different letters indicate significantly different values at *p* < 0.05.

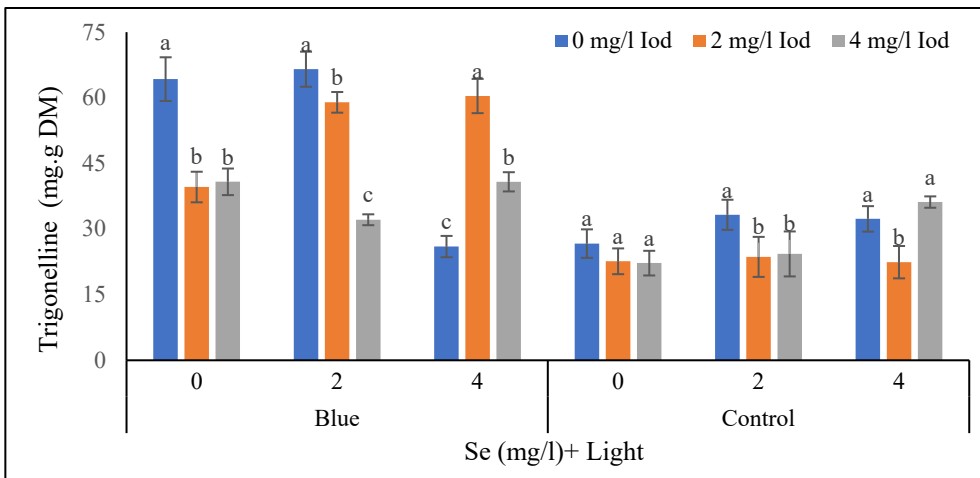

**Figure 10.** Mean comparison of selenium, I and light combined effects on trigonelline in fenugreek shoots in 80 days after seed planting. Different letters indicate significantly different values at $p < 0.05$.

### 3.3.2. Seed Trigonelline Content

As presented in Figure 11, in the vegetative stage, in the supplemented blue lighting with the 0 mg $L^{-1}$ Se treatment, the highest and the lowest seed trigonelline contents were noticed in 4 and 0 mg $L^{-1}$ of I, respectively. No significant difference was detected between 0 and 2 mg $L^{-1}$ of I. In the supplemented blue light and 4 mg $L^{-1}$ of Se, the highest and the lowest seed trigonelline content was achieved in 0 and 2 mg $L^{-1}$ of I, respectively. In non-supplemented blue light conditions and 0 mg $L^{-1}$ of Se, different results arose. Under the blue lighting treatment and 2 mg $L^{-1}$ of Se, the highest and the lowest seed trigonelline content was recorded in 2 and 0 mg $L^{-1}$ of I, respectively. In the non-supplemented blue light conditions, the highest and the lowest trigonelline content was noted in 4 and 0 mg $L^{-1}$ of I, respectively (Figure 11).

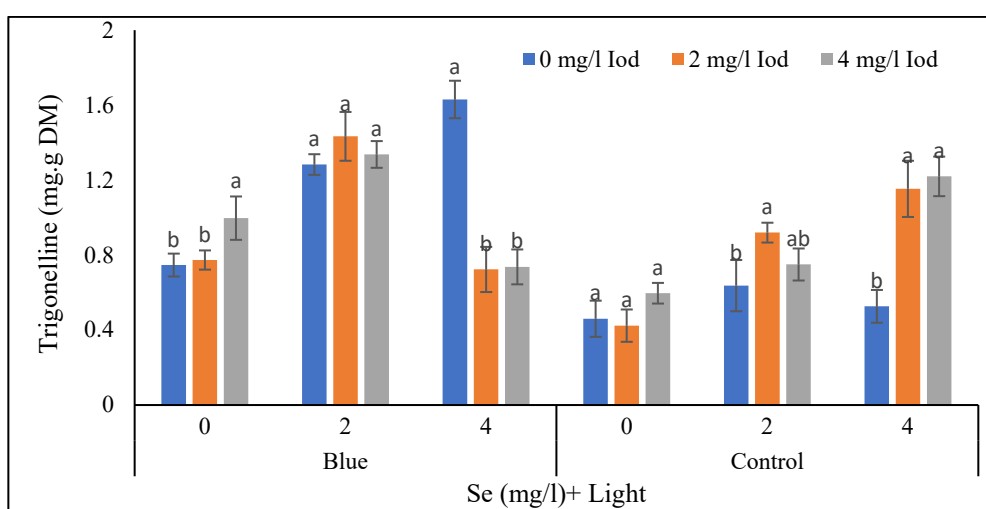

**Figure 11.** Mean comparison of selenium, I and light combined effects on seed trigonelline in fenugreek plant. Different letters indicate significantly different values at $p < 0.05$.

## 4. Discussion

It seems that blue light supplementation inhibits plant length but increases biomass. He et al. [23] reported that the total dry weight of edible broccoli sprouts increased significantly under the Se application and high fraction of blue light, so that the highest dry weight of broccoli (edible sprouts) was achieved in 100 μmol $L^{-1}$ of Se combined with red + blue treatment (1:2) [23]. Blue light increases the photosynthetic capacity of a plant

and consequently enhances plant growth [34]. It was found that in the hydroponic production of cabbage, incorporating I into the nutrient solution increases the dry weight of plant tissues [28], as was noticed in this study. In coriander (*Coriandrum sativum*), a higher dry and fresh mass accumulation was observed in different ratios of red and blue light compared to plants grown under 100% red light [40]. It seems that the blue light affecting stomatal cells causes an improved absorption pressure in the vessels to move more water from the roots to the leaves and this makes an increased RWC in the leaf, as we observed in this experiment. In fact, with the activation of the stomatal guard cells of the leaf, the transpiration rate will be increased and this in turn improves the flow of water tension in the xylem, and water and nutrient absorption.

The highest amounts of total leaf chlorophyll in both the 40 and 80 days after sowing were observed in the interaction of the supplemented blue light with 4 mg $L^{-1}$ of Se. It can be argued that the increase in dry weight under supplementary blue light is correlated with the increase in chlorophyll content. This is consistent with the remarks of Sæbø et al. [41], who found that in *Betula pendula* leaves, the chlorophyll amount was nearly two-fold as high under blue light compared to white and red light. This may be because under the blue lighting, the net area of chloroplasts in each cell of the leaves was the largest and the starch accumulated in such cells was less than in cells treated with red light. Therefore, the inhibition of chlorophyll biosynthesis was less, leading to a higher chlorophyll concentration.

In line with our study, previous literature presents that total chlorophyll in lettuce [24] and broccoli [25] increased under Se treatment (blue). Moreover, in current research, the highest anthocyanin content in both developmental stages is related to blue light supplementations and 4 mg $L^{-1}$ of Se. At the pod formation stage under the blue light supplementation, the maximum carotenoids were accumulated in 4 mg $L^{-1}$ of I. This finding is inconsistent with the findings of Duborská et al. [42], who reported that carotenoid and chlorophyll content decreased due to iodine application. Blue light can activate the expression of genes involved in the production of enzymes such as PAL (phenylalanine ammonia lyase), CHS (chalcone synthase) and DFR (dihydroflavonol-4-reductase), which are key elements in the biosynthetic pathways of anthocyanins [43]. In fact, stimulating the photoreceptors with a higher fraction of blue light results in more signaling in the plant, and the genes that are involved in the production of enzymes responsible for the biosynthesis of anthocyanins will increase [44]. The application of 60 and 120 mg $L^{-1}$ of Se reduced the chlorophyll content of mint while increasing the carotenoids [26].

Generally, in the current research, the trigonelline contents in the fenugreek shoot and seed are highly variable under light, Se and I treatments, and also developmental stages. It has been documented that blue light can drive the expression of genes involved in the production of enzymes that are key elements in the biosynthetic pathways of secondary metabolism [43]. In line with our study, reports show that blue light is effective in increasing the level of phenolic compounds [45], ascorbic acid [46], carotenoids [47] and anthocyanins [48], and as a result, can increase the biochemical quality of vegetables.

Under supplementary blue light treatment with the application of 4 mg $L^{-1}$ of Se, the highest seed yield per plant was observed in the 0 mg $L^{-1}$ I application. Cunha et al. [49] reported that in response to Se fertilization, the concentration of photosynthetic pigments, the activity of antioxidant enzymes and the concentration of total sugar in *Arachis hypogaea* L. leaves increased. In addition, Se increases the activity of nitrate reductase, which leads to a higher concentration of ureides, amino acids and proteins, and improves nitrogen assimilation efficiency. Therefore, it can be justified in this way that one of the reasons for the improvement in seed yield due to the application of selenium is the improvement in nitrogen assimilation efficiency and the increase in photosynthetic pigments, as, in this study, the highest chlorophyll content was observed in 4 mg $L^{-1}$ of Se.

## 5. Conclusions

This study was conducted with the aim of evaluating the effects of supplementary blue light and fertilization with iodine and selenium to improve the quantitative and qualitative performance of fenugreek under greenhouse conditions. The results showed that the addition of 15 $\mu$mol m$^{-2}$ s$^{-1}$ is more suitable for improving photosynthetic pigments, the relative water content of the leaf, growth characteristics, seed yield and trigonelline content in fenugreek. Considering that the use of supplementary light is an eco-friendly method, it is recommended to use it in a greenhouse (2 h in the morning) during fenugreek cultivation. The results also showed that the application of 4 mg L$^{-1}$ of selenium can improve trigonelline content and seed yield in fenugreek. It also seems that the application of iodine was able to develop the root growth; however, we need studies to evaluate the effects of iodine on root growth and root architecture. Also, in the future, further research with other lighting parameters and time periods of using supplementary lighting is suggested to optimize other parameters of lighting and the effective time of using supplementary blue light.

**Author Contributions:** Conceptualization, M.Z., Y.F. and D.R.; methodology, S.R., B.Y. and D.R.; investigation, B.Y. and E.P.; writing—original draft preparation, S.R. and B.Y.; writing—review and editing, Y.F., B.Y. and M.Z.; funding acquisition, M.Z., E.P. and D.R. All authors have read and agreed to the published version of the manuscript.

**Funding:** This research was funded by the RUDN University Scientific Grant System (project № 202193-2-000).

**Data Availability Statement:** Not applicable.

**Acknowledgments:** The authors are thankful to the University of Zabol and RUDN University for laboratory services and funding acquisition.

**Conflicts of Interest:** The authors declare no conflict of interest.

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
