# Peer review of "Selenium and Iodine Biofortification Interacting with Supplementary Blue Light to Enhance the Growth Characteristics, Pigments, Trigonelline and Seed Yield of Fenugreek (Trigonella foenum-gracum L.)"

_agronomy, doi:10.3390/agronomy13082070_

Round 1

Reviewer 1 Report

pragraph :Combined effects of light, selenium and iodine on fenugreek plant pigments from row 300 to 314 

this section must include recent referenchere and the discussion sould be enriched based on recent references such as the following that they bring more interesting information about the effect of light on medecinal metabolite responsible genes activation 

1)      Blue light-mediated transcriptional activation and repression of gene expression in bacteria Premkumar Jayaraman, Kavya Devarajan, Tze Kwang Chua, Hanzhong Zhang, Erry Gunawan, Chueh Loo PohNucleic Acids Research, Volume 44, Issue 14, 19 August 2016, Pages 6994–7005, https://doi.org/10.1093/nar/gkw548

2)       w.frontiersin.org/articles/10.3389/fpls.2021.781236/full

3)       https://doi.org/10.3389/fpls.2021.781236

4)      https://link.springer.com/chapter/10.1007/978-981-15-1761-7_3 

a question to authors : 

please try to refer to AOAC methods standards for the  measurment of secondary metabolites or spectroscopic methods

as Agronomy is a Journal that brings useful information and practical solutions to field :  Funegrec is a field crop do you think using blue light or such solution is usefull to crops thats requires space and big fields , if such technic is for pharmaceutical use wht will be the economic benefit compared to other technics used for bifortification of trigonella such as biostimulants ?

Author Response

Dear reviewer 

We gratefully acknowledge the detailed revision of the text and useful suggestions to improve the paper by the reviewer. We have closely followed he/she suggestions and introduced the required changes in the text. Main changes are performed through Track changes function

Reviewers' comment: pragraph :Combined effects of light, selenium and iodine on fenugreek plant pigments from row 300 to 314 

this section must include recent referenchere and the discussion sould be enriched based on recent references such as the following that they bring more interesting information about the effect of light on medecinal metabolite responsible genes activation 

1)      Blue light-mediated transcriptional activation and repression of gene expression in bacteria Premkumar Jayaraman, Kavya Devarajan, Tze Kwang Chua, Hanzhong Zhang, Erry Gunawan, Chueh Loo PohNucleic Acids Research, Volume 44, Issue 14, 19 August 2016, Pages 6994–7005, https://doi.org/10.1093/nar/gkw548

2)       w.frontiersin.org/articles/10.3389/fpls.2021.781236/full

3)       https://doi.org/10.3389/fpls.2021.781236

4)      https://link.springer.com/chapter/10.1007/978-981-15-1761-7_3 

Authors' comment: Thanks a lot for the suggestion. We tried to increase the use of recent references in the revised version.

Reviewers' comment: a question to authors : please try to refer to AOAC methods standards for the measurment of secondary metabolites or spectroscopic methods.

Authors' comment: We have not had access to these methods, however, we have tried to use methods taken from articles published in reliable scientific journals.

Reviewers' comment: as Agronomy is a Journal that brings useful information and practical solutions to field :  Funegrec is a field crop do you think using blue light or such solution is usefull to crops thats requires space and big fields , if such technic is for pharmaceutical use wht will be the economic benefit compared to other technics used for bifortification of trigonella such as biostimulants ?

Authors' comment: Thank you for providing this valuable comment. Biodiversity in greenhouses is an important issue for income growth, efficient use of natural resources and increased resource productivity (e.g. land, labor, water, fertilizer and energy), sustainable development and ecological management/improvement. Considering that it takes about 6 weeks from planting seeds to flowering of fenugreek, and this time can be reduced during greenhouse cultivation. Therefore, with the aim of using fenugreek tissue, it is recommended to grow it in a greenhouse to improve biodiversity due to its relatively short growth period. Considering that various studies have shown the improvement of the nutritional value of leafy vegetables under blue light, for this reason we have been searching for a way to increase the qualitative performance of fenugreek without reducing its quantitative performance.

Reviewer 2 Report

The manuscript on the ‘Assessment of selenium, iodine and supplementary blue light effects on the growth characteristics, pigments, trigonelline and seed yield of fenugreek (Trigonella foenum-gracum L.)’ is of interest regarding biofortification.

Abstract

It is not organized scientifically. Rewrite it and improve the English language (see attachment). Add the name of growth stages rather than days throughout the manuscript.

Introduction

Introduction section is well managed and needs little bit improvement regarding language.

Material and methods

This section is poorly presented. It needs extensive improvements.

Result

Improve this section with interpretation of significant findings. It looks too lengthy.

Discussion

Improve this section with logical and scientific approach. Rewriting of previous studies/results deteriorate the quality of manuscript. No need of subheadings in discussion section.

Conclusion

Conclusion may be short, specific and quantified.

Reference

The authors need to work on the reference list according to the journal style and cross matching.

Figures

Merge the figures with each other having from same aspect. Figure quality may be improved.

For further details, find the attachment please.

Please invest time and energy to improve the English language. It looks that write up is very poor.

Author Response

Dear reviewer 

We gratefully acknowledge the detailed revision of the text and useful suggestions to improve the paper by the reviewer. We have closely followed he/she suggestions and introduced the required changes in the text. Main changes are performed through Track changes function

The manuscript on the ‘Assessment of selenium, iodine and supplementary blue light effects on the growth characteristics, pigments, trigonelline and seed yield of fenugreek (Trigonella foenum-gracum L.)’ is of interest regarding biofortification.

Reviewers' comment: Abstract: It is not organized scientifically. Rewrite it and improve the English language (see attachment). Add the name of growth stages rather than days throughout the manuscript.

Authors' comment: We appreciate Reviewer for this valuable comment, the abstract was reviewed. Throughout the text growth stages were considered instead of number of days. Native English editing was also considered throughout the text.

Reviewers' comment: Introduction: Introduction section is well managed and needs little bit improvement regarding language.

Authors' comment: Native English editing was considered throughout the text.

Reviewers' comment: Material and methods: This section is poorly presented. It needs extensive improvements.

Authors' comment: Details were added to the Materials and Methods, which are now available.

Reviewers' comment: Result: Improve this section with interpretation of significant findings. It looks too lengthy.

Authors' comment: We tried to reduce the volume of results and focus on important findings and their interpretation.

Reviewers' comment: Discussion: Improve this section with logical and scientific approach. Rewriting of previous studies/results deteriorate the quality of manuscript. No need of subheadings in discussion section.

Authors' comment: Thanks for this remark. We have reorganized the discussion section and removed the subheadings in discussion section.

Reviewers' comment: Conclusion: Conclusion may be short, specific and quantified.

Authors' comment: Revised it.

Reviewers' comment: Reference: The authors need to work on the reference list according to the journal style and cross matching.

Authors' comment: Reference list modified based on Instructions for Authors.

Reviewers' comment: Figures: Merge the figures with each other having from same aspect. Figure quality may be improved.

For further details, find the attachment please.

Authors' comment: We merged figures based on same aspect and tried to increase the quality of figures.

Reviewers' comment: Comments on the Quality of English Language: Please invest time and energy to improve the English language. It looks that write up is very poor. 

Authors' comment: Native English editing was considered throughout the text

Reviewer 3 Report

Comments for  manuscript jof-2480390

The main purpose of the study was to investigate the effects of selenium, iodine and blue light on growth, biomass weight, physicochemical properties, mineral elements and the medicinal composition of trigonelline in fenugreek seeds and shoots at the 40 and 80-day stage. According to the authors, vegetable biofortification is a reliable method of increasing dietary intake of iodine and selenium.

Abstract: it is exceeds 200 words according to journal instructions, so I would recommend the authors to make it short and more precise, and follow the style of structured abstracts.

Introduction; it covered the research point, provides some details of research problem, but authors do not address anything related to the importance of the model plant they used to carry out their research and the reasons why they selected it.

Materials and Methods: the section is too long, it is suggested to reduce the description, and in some parts, such as line 102, authors should explain what they mean by “fresh weight of shoot, root and seed was measured with a digital scale”. It is not appropriate to mix unit symbols with unit names in the same expression, such as line 127 and 128 “One g of the plant samples (leaf or seed) was mixed with one g of magnesium oxide”

Results: it is too long and it can be repetitive and confusing, it is suggested to improve the quality of writing.

Discussion: the discussion should be supported and justified accordingly with some latest references. The authors do not interpret them results in perspective of previous studies, the ideas do not connect with each other. For example, in lines 312-314 the authors discuss something other than what is presented in the subtitle and that happens in various sections of the discussion.

Most of the literature is cited correctly, but some references are presented with different formats. It is recommended to prepare the references according to the journal indications, using a bibliography software package.

Considering the quality of the manuscript, it is suggested to rewrite and synthesize the Materials and methods  and Discussion section in a way that reflects the relevance of the work.

Author Response

Dear reviewer 

We gratefully acknowledge the detailed revision of the text and useful suggestions to improve the paper by the reviewer. We have closely followed he/she suggestions and introduced the required changes in the text. Main changes are performed through Track changes function

The main purpose of the study was to investigate the effects of selenium, iodine and blue light on growth, biomass weight, physicochemical properties, mineral elements and the medicinal composition of trigonelline in fenugreek seeds and shoots at the 40 and 80-day stage. According to the authors, vegetable biofortification is a reliable method of increasing dietary intake of iodine and selenium.

Reviewers' comment: Abstract: it is exceeds 200 words according to journal instructions, so I would recommend the authors to make it short and more precise, and follow the style of structured abstracts.

Authors' comment: According to the reviewer's opinion, the abstract was revised.

Reviewers' comment: Introduction; it covered the research point, provides some details of research problem, but authors do not address anything related to the importance of the model plant they used to carry out their research and the reasons why they selected it.

Authors' comment: Accordingly, some explanations were added. The most important importance of fenugreek plant is to improve insulin function.

Reviewers' comment: Materials and Methods: the section is too long, it is suggested to reduce the description, and in some parts, such as line 102, authors should explain what they mean by “fresh weight of shoot, root and seed was measured with a digital scale”. It is not appropriate to mix unit symbols with unit names in the same expression, such as line 127 and 128 “One g of the plant samples (leaf or seed) was mixed with one g of magnesium oxide”

Authors' comment: It was modified.

Reviewers' comment: Results: it is too long and it can be repetitive and confusing, it is suggested to improve the quality of writing.

Authors' comment: It was modified.

Reviewers' comment: Discussion: the discussion should be supported and justified accordingly with some latest references. The authors do not interpret them results in perspective of previous studies, the ideas do not connect with each other. For example, in lines 312-314 the authors discuss something other than what is presented in the subtitle and that happens in various sections of the discussion.

Authors' comment: Thanks for this remark. We have reorganized the discussion section.

Most of the literature is cited correctly, but some references are presented with different formats. It is recommended to prepare the references according to the journal indications, using a bibliography software package.

Authors' comment: We thank the Reviewer for this comment, we checked the references and made changes according to the journal format.

Considering the quality of the manuscript, it is suggested to rewrite and synthesize the Materials and methods and Discussion section in a way that reflects the relevance of the work.

Authors' comment: Materials and methods and Discussion section were revised and we think it is now significantly improved.

Reviewer 4 Report

The article 'Assessment of selenium, iodine and supplementary blue light effects on the growth characteristics, pigments, trigonelline and seed yield of fenugreek (Trigonella foenum-gracum L.) - is interesting. It fits in well with the current research issues concerning biofortification. I have no doubt that the research carried out is of cognitive and utilitarian value. In the pot studies, the workload is certainly less than in the long-term field experiments, but on the other hand, only in the cover crop was it possible to control the light.

The individual chapters need to be supplemented and the issues mentioned in the review clarified. Results generally presented in a clear manner, although a more insightful interpretation could have been made as in the discussion. Detailed notes and comments are provided below. The Authors should take these into account when preparing the manuscript for publication.

Keywords: Biofortification; Supplementary blue light; Growth; Health-preserving compounds; Trigonelline; Micronutrients – should be lower case - biofortification; supplementary blue light….

1.            Introduction

Cited items of literature should be assigned numbers, e.g.:

Line 33 - should be 1. Introduction

Line 34 - is (Schomburg and Köhrle, 2008) - should be - [1]

The list of references should be in the order of citations, not alphabetically.

Adapt to the requirements of the journal Agronomy!

Introduction, should necessarily include information about Trigonella foenum-gracum L., which determined the choice of this plant for the experiment. In most European countries, it is an herbaceous plant, so its consumption is limited to medicinal purposes. As a fodder plant and vegetable it is cultivated in Eastern Europe and Asia. Please, for the sake of clarity, give an introduction to this plant based on the literature.

Lines -83-85 - The research objective should be more precise.

2.            Materials and Methods

Line 88 - there is "a three-factor factorial experiment" - it should be - three-factor experiment

Line 91 - there is "two supplement blue light levels" - probably not needed in the record "blue". There is one blue light source and control.

Line 92 - Was the seed certified? Why were the seeds washed in dishwashing liquid?

Lines 92-93 - what was the capacity of the pots? How many pots were there in total in the experiment?

Lines 101-105 - how many plants were in each combination? on how many plants were the analyzes performed?

Line 123 - how large was the sample, please specify.

Line 133 - what units are the weight of the seeds in?

Lines 135-137 - the statistical program is one thing, but what statistical analysis was done? For three factors? Or was each factor analyzed separately? The relationships could be determined by correlation or PCA (principal component analysis). This would help to better describe the results, or express regularities between the variables. I recommend authors to study statistical analysis in the future.

This note applies to all tables and figures!

3. Results (number the sections and subsections as required by Agronomy).

This chapter requires a more thorough analysis. Tables and figures had to be included in the text, the lack of them significantly hinders the analysis of the results.

The names of tables and figures should be simplified, it is obvious that the tables compare the results, and the footnotes under the tables explain, for example, the averages of the procedure

Table 1. Means comparison of Effect of selenium on carotenoid, anthocyanin, plant height, fresh weight of shoot, chlorophylll and RWC in fenugreek plant.

Table 4. Means comparison of Effect of light and selenium combined effects on plant height, chlorophyll and RWC in fenugreek in 40 days after seed planting.

4. Discussion

This chapter lacks specific literature references to the results of the research conducted by the Authors. (Own research should largely correspond to the literature). In its current form, the discussion contains the content referred to in the 1st Introduction.

5. Conclusions

Why were these studies conducted? What has been achieved? Some practical tips. Prospects of use, direction of further research.

References

Lines 404 and 408 - same position

The position of Mizuno et al. 2015 quoted in the Discussion is missing (in references is from 2011)

Author Response

Dear reviewer 

We gratefully acknowledge the detailed revision of the text and useful suggestions to improve the paper by the reviewer. We have closely followed he/she suggestions and introduced the required changes in the text. Main changes are performed through Track changes function

Reviewers' comment: The article 'Assessment of selenium, iodine and supplementary blue light effects on the growth characteristics, pigments, trigonelline and seed yield of fenugreek (Trigonella foenum-gracum L.) - is interesting. It fits in well with the current research issues concerning biofortification. I have no doubt that the research carried out is of cognitive and utilitarian value. In the pot studies, the workload is certainly less than in the long-term field experiments, but on the other hand, only in the cover crop was it possible to control the light.

The individual chapters need to be supplemented and the issues mentioned in the review clarified. Results generally presented in a clear manner, although a more insightful interpretation could have been made as in the discussion. Detailed notes and comments are provided below. The Authors should take these into account when preparing the manuscript for publication.

Authors' comment: The comment of the respected Reviewer is very encouraging and we will try to apply all the comments in the manuscript to improve the level of work.

Reviewers' comment: Keywords: Biofortification; Supplementary blue light; Growth; Health-preserving compounds; Trigonelline; Micronutrients – should be lower case - biofortification; supplementary blue light….

Authors' comment: It was done.

Reviewers' comment: 1. Introduction

Cited items of literature should be assigned numbers, e.g.:

Line 33 - should be 1. Introduction

Line 34 - is (Schomburg and Köhrle, 2008) - should be - [1]

The list of references should be in the order of citations, not alphabetically.

Adapt to the requirements of the journal Agronomy!

Authors' comment: It was done based on Agronomy format.

Introduction, should necessarily include information about Trigonella foenum-gracum L., which determined the choice of this plant for the experiment. In most European countries, it is an herbaceous plant, so its consumption is limited to medicinal purposes. As a fodder plant and vegetable, it is cultivated in Eastern Europe and Asia. Please, for the sake of clarity, give an introduction to this plant based on the literature.

Authors' comment: Accordingly, some explanations were added.

Lines -83-85 - The research objective should be more precise.

Authors' comment: We tried to rewrite the purpose of the study more clearly.

Reviewers' comment: 2. Materials and Methods

Line 88 - there is "a three-factor factorial experiment" - it should be - three-factor experiment

Authors' comment: It was corrected.

Line 91 - there is "two supplement blue light levels" - probably not needed in the record "blue". There is one blue light source and control.

Authors' comment: It was corrected.

Line 92 - Was the seed certified? Why were the seeds washed in dishwashing liquid?

Authors' comment: Regarding the seeds, an explanation was added in the text of the Materials and Methods section, which is now available.

Lines 92-93 - what was the capacity of the pots? How many pots were there in total in the experiment?

Authors' comment: Added the pots features to the Materials and Methods section, which is now available.

Lines 101-105 - how many plants were in each combination? on how many plants were the analyzes performed?

Authors' comment: Some explanation was added to the text.

Line 123 - how large was the sample, please specify.

Authors' comment: Some explanation was added to the text.

Line 133 - what units are the weight of the seeds in?

Authors' comment: Some explanation was added to the text.

Lines 135-137 - the statistical program is one thing, but what statistical analysis was done? For three factors? Or was each factor analyzed separately? The relationships could be determined by

correlation or PCA (principal component analysis). This would help to better describe the results, or express regularities between the variables. I recommend authors to study statistical analysis in the future.

This note applies to all tables and figures!

Authors' comment: It

Reviewers' comment: 3. Results (number the sections and subsections as required by Agronomy).

This chapter requires a more thorough analysis. Tables and figures had to be included in the text, the lack of them significantly hinders the analysis of the results.

The names of tables and figures should be simplified, it is obvious that the tables compare the results, and the footnotes under the tables explain, for example, the averages of the procedure

Table 1. Means comparison of Effect of selenium on carotenoid, anthocyanin, plant height, fresh weight of shoot, chlorophylll and RWC in fenugreek plant.

Table 4. Means comparison of Effect of light and selenium combined effects on plant height, chlorophyll and RWC in fenugreek in 40 days after seed planting.

Authors' comment: All these comments were applied in the text.

Reviewers' comment: 4. Discussion

This chapter lacks specific literature references to the results of the research conducted by the Authors. (Own research should largely correspond to the literature). In its current form, the discussion contains the content referred to in the 1st Introduction.

Authors' comment: Thanks for this remark. We have reorganized the discussion section.

Reviewers' comment: 5. Conclusions

Why were these studies conducted? What has been achieved? Some practical tips. Prospects of use, direction of further research.

Authors' comment: Based on the comment of the respected Reviewer, the conclusion was completed.

Reviewers' comment: References

Lines 404 and 408 - same position

The position of Mizuno et al. 2015 quoted in the Discussion is missing (in references is from 2011)

Authors' comment: It was modified.

Round 2

Reviewer 2 Report

Clean copy contains a lot of mistakes.

Quality of English Language is improved by the authors.

Author Response

Dear reviewer 

We found the mistakes and revised throughout the text.

Regards,

Meisam

Reviewer 3 Report

I would like to thank the authors for making the suggested changes.

Author Response

Thanks dear reviewer